

# Glyoxal yield from isoprene oxidation and relation to formaldehyde: chemical mechanism, constraints from SENEX aircraft observations, and interpretation of OMI satellite data

Christopher Chan Miller[1], Daniel J. Jacob[1,2], Eloise A. Marais[1], Karen Yu[2], Katherine R. Travis[2], Patrick S. Kim[1], Jenny A. Fisher[3], Lei Zhu[2], Glenn M. Wolfe[4,5], Frank N. Keutsch[6], Jennifer Kaiser[6,a], Kyung-Eun Min[7,8,b], Steven S. Brown[8,9], Rebecca A. Washenfelder[7,8], Gonzalo González Abad[10], and Kelly Chance[10]

[1]Department of Earth and Planetary Sciences, Harvard University, Cambridge, MA, USA
[2]School of Engineering and Applied Sciences, Harvard University, Cambridge, MA, USA
[3]School of Chemistry and School of Earth and Environmental Sciences, University of Wollongong, Wollongong, NSW, Australia
[4]Atmospheric Chemistry and Dynamics Lab, NASA Goddard Space Flight Center, Greenbelt, MD, USA
[5]Joint Center for Earth Systems Technology, University of Maryland Baltimore County, Baltimore, MD, USA
[6]Department of Chemistry, University of Wisconsin Madison, Madison, WI, USA
[7]Cooperative Institute for Research in Environmental Sciences, University of Colorado Boulder, Boulder, CO, USA
[8]Chemical Sciences Division, NOAA Earth System Research Laboratory, Boulder, CO, USA
[9]Department of Chemistry and Biochemistry, University of Colorado, Boulder, CO, USA
[10]Harvard-Smithsonian Center for Astrophysics, Cambridge MA, USA
[a]now at: School of Engineering and Applied Sciences, Harvard University, Cambridge MA, USA
[b]now at: School of Earth Sciences and Environmental Engineering, Gwangju Institute of Science and Technology, Gwangju, South Korea

*Correspondence to:* Daniel Jacob (djacob@fas.harvard.edu)

**Abstract.**

Glyoxal (CHOCHO) is produced in the atmosphere by oxidation of volatile organic compounds (VOCs). It is measurable from space by solar backscatter along with formaldehyde (HCHO), another oxidation product of VOCs. Isoprene emitted by vegetation is the dominant source of CHOCHO and HCHO in most of the world. We use aircraft observations of CHOCHO and HCHO from the SENEX campaign over the Southeast US in summer 2013 to better understand the time-dependent yields from isoprene oxidation, their dependences on nitrogen oxides ($NO_x \equiv NO + NO_2$), the behaviour of the CHOCHO-HCHO relationship, the quality of OMI satellite observations, and the implications for using satellite CHOCHO observations as constraints on isoprene emission. We simulate the SENEX and OMI observations with the GEOS-Chem chemical transport model featuring a new chemical mechanism for CHOCHO formation from isoprene. The mechanism includes prompt CHOCHO formation under low-$NO_x$ conditions following the isomerization of the isoprene peroxy radical ($ISOPO_2$). The SENEX observations provide support for this prompt CHOCHO formation pathway, and are generally consistent with the GEOS-Chem mechanism. Boundary layer CHOCHO and HCHO are strongly correlated in the observations and the model, with some departure under low-$NO_x$ conditions due to prompt CHOCHO formation. SENEX vertical profiles indicate a free tropospheric CHOCHO background that is absent from the model. The OMI CHOCHO data provide some support for this free tropospheric





background and show Southeast US enhancements consistent with the isoprene source but a factor of 2 too low. Part of this OMI bias is due to excessive surface reflectivities assumed in the retrieval. The OMI CHOCHO and HCHO seasonal data over the Southeast US are tightly correlated and provide redundant proxies of isoprene emission. Higher temporal resolution in future geostationary satellite observations may enable detection of the prompt CHOCHO production under low-$NO_x$ conditions
apparent in the SENEX data.

## 1  Introduction

Glyoxal (CHOCHO) and formaldehyde (HCHO) are short-lived products of the atmospheric oxidation of volatile organic compounds (VOCs). Both are detectable from space by solar backscatter (Chance et al., 2000; Wittrock et al., 2006). Isoprene emitted by terrestrial vegetation accounts for about a third of the global source of non-methane VOCs (NMVOCs) (Guenther
et al., 2012) and drives large enhancements of CHOCHO and HCHO in the continental boundary layer (Palmer et al., 2003; Fu et al., 2008). Satellite observations of HCHO have been widely used as a proxy to estimate isoprene emission (Abbot et al., 2003; Palmer et al., 2006; Millet et al., 2008; Curci et al., 2010; Barkley et al., 2013), but there are uncertainties related to the HCHO yield from isoprene oxidation (Marais et al., 2012) and the role of other NMVOCs as HCHO precursors (Fu et al., 2007). CHOCHO observations from space could provide a complementary constraint (Vrekoussis et al., 2009, 2010; Alvarado
et al., 2014; Chan Miller et al., 2014). Here we use CHOCHO and HCHO aircraft observations over the Southeast United States from the Summer 2013 Southeast Nexus (SENEX) campaign (Warneke et al., 2016), interpreted with the GEOS-Chem chemical transport model (CTM), to test understanding of the CHOCHO yield from isoprene oxidation, its dependence on nitrogen oxide radicals ($NO_x \equiv NO + NO_2$), and the combined value of the HCHO-CHOCHO pair measured from space to constrain isoprene emissions and chemistry.

Isoprene impacts air quality and climate as a precursor to ozone (Geng et al., 2011) and secondary organic aerosol (SOA) (Carlton et al., 2009), and also affects concentrations of hydrogen oxide radicals ($HO_x \equiv OH+$peroxy radicals) (Peeters and Muller, 2010) and $NO_x$ (Mao et al., 2013; Fisher et al., 2016). Atmospheric oxidation of isoprene by OH takes place on a timescale of less than an hour to produce organic peroxy radicals ($ISOPO_2$). Reaction of $ISOPO_2$ with NO drives production of ozone and of organic nitrates that serve as a reservoir for $NO_x$ (Browne and Cohen, 2012). At lower $NO_x$ levels,
$ISOPO_2$ reacts dominantly with $HO_2$ to produce isoprene epoxydiols (IEPOX) via isoprene peroxides (ISOPOOH) (Paulot et al., 2009b), and from there isoprene SOA (Marais et al., 2016). $ISOPO_2$ can also isomerize to generate $HO_x$ radicals (Peeters et al., 2009; Crounse et al., 2011; Peeters et al., 2014).

The fate of $ISOPO_2$ determines the production rates and overall yields of CHOCHO and HCHO. Several studies have provided insight on the time- and $NO_x$-dependent yield of HCHO (Palmer et al., 2003; Marais et al., 2012; Wolfe et al., 2016).
Under high-$NO_x$ conditions, HCHO production is sufficiently prompt that observed HCHO columns can be locally related to isoprene emission rates (Palmer et al., 2006). This assumption is the basis of many studies that have used satellite HCHO observations to constrain isoprene emissions (Palmer et al., 2006; Fu et al., 2007; Millet et al., 2008; Curci et al., 2010). HCHO production is much slower under low-$NO_x$ conditions, spatially "smearing" the local relationship between isoprene emissions



and HCHO columns. This has been addressed by using concurrent satellite data for $NO_2$ columns to correct the isoprene-HCHO relationship (Marais et al., 2012) or by using adjoint-based inverse modeling to relate HCHO columns to isoprene emissions including the effect of transport (Fortems-Cheiney et al., 2012).

Isoprene is estimated to account for about $\sim 50\%$ of global CHOCHO production (Fu et al., 2008), but there is large
uncertainty regarding the yield. Open fires and aromatic VOCs also produce CHOCHO with high yield (Volkamer et al., 2001; Fu et al., 2008; Chan Miller et al., 2016). Several studies have used the measured CHOCHO-HCHO concentration ratio $R_{GF} = [CHOCHO]/[HCHO]$ as an indicator of the dominant VOC precursors. Vrekoussis et al. (2010) found higher $R_{GF}$ values ($> 0.04$) from GOME-2 satellite observations in regions where biogenic VOCs are dominant, and lower values where anthropogenic VOCs are dominant. However, the opposite behaviour is observed from ground-based studies (DiGangi et al.,
2012). Our recent CHOCHO retrieval from the OMI satellite instrument (Chan Miller et al., 2014) is in better agreement with surface observations of CHOCHO and $R_{GF}$ (Kaiser et al., 2015) as a result of improved background corrections and removal of $NO_2$ interferences. There remains the question of how observed CHOCHO-HCHO relationships are to be interpreted.

The Southeast Nexus (SENEX) aircraft campaign was conducted over the Southeast United States in June-July 2013. The aircraft had a detailed chemical payload including in situ CHOCHO (Min et al., 2016) and HCHO (Cazorla et al., 2015).
Thirteen daytime flights were conducted over the campaign with extensive boundary layer coverage. A previous comparison of the SENEX observations to AM3 CTM simulations highlighted the CHOCHO yield uncertainty in current isoprene oxidation mechanisms (Li et al., 2016). Here we present an improved chemical mechanism for CHOCHO formation from isoprene for the GEOS-Chem CTM. We use the SENEX observations to evaluate the CHOCHO formation pathways from isoprene in the new mechanism and in the Master Chemical Mechanism v3.3.1 (Jenkin et al., 2015). Wolfe et al. (2016) used their SENEX
HCHO observations to analyze the HCHO yield from isoprene and its time- and $NO_x$-dependence. We apply here some of the same methods to analyze the CHOCHO yield.

## 2 GEOS-Chem Model Description

### 2.1 General Description

We use the same version of GEOS-Chem v9.2 (http://www.geos-chem.org) that has been used previously to interpret obser-
25 vations from the NASA SEAC⁴RS aircraft campaign conducted in the same Southeast US region in August-September 2013 (Toon et al., 2016). The model is driven by assimilated meteorological data with $0.25° \times 0.3125°$ horizontal resolution from the Goddard Earth Observing System (GEOS-FP) reanalysis product (Molod et al., 2012). The native $0.25° \times 0.3125°$ resolution is retained in GEOS-Chem over the North American domain ($130° - 60°$W, $9.75° - 60°$N ), nested within a global simulation at $2° \times 2.5°$ resolution (Kim et al., 2015). Isoprene chemistry in v9.2 is as described by Mao et al. (2013), but the SEAC⁴RS
simulation includes a number of updates described by Travis et al. (2016) and Fisher et al. (2016). The simulation presented here includes further modifications relevant to CHOCHO, listed in the supplementary material (Table S1), and summarized below. Evaluation of the model with SEAC⁴RS observations has been presented by Kim et al. (2015) for aerosols, Travis et al.





(2016) for ozone and $NO_x$, Fisher et al. (2016) for organic nitrates, Marais et al. (2016) for isoprene SOA, and Zhu et al. (2016) for HCHO including satellite validation.

Isoprene emissions in the model are from MEGANv2.1 (Guenther et al., 2012) with a 15% reduction (Kim et al., 2015), and $NO_x$ emissions are as described by Travis et al. (2016) including a 50% decrease in the anthropogenic source relative to the

2013 National Emission Inventory of the U.S. Environmental Protection Agency. Yu et al. (2016) pointed out that isoprene and $NO_x$ emissions in the Southeast US are spatially segregated and show that the $0.25° \times 0.3125°$ resolution of GEOS-Chem is adequate for separating the populations of high- and low-$NO_x$ conditions for isoprene oxidation.

## 2.2  CHOCHO Formation from Isoprene and Loss Pathways

Figure 1 shows the CHOCHO formation pathways from isoprene oxidation by OH (the main isoprene sink) as implemented

in this work. Oxidation is initiated by OH addition to the terminal carbons of the isoprene double bonds (positions 1 and 4, Figure 1). Isoprene peroxy radicals (ISOPO$_2$) are formed by $O_2$ addition to the carbon either in $\beta$ or $\delta$ to the hydroxyl carbon. ISOPO$_2$ reacts with NO and HO$_2$, and also isomerizes. Together these pathways represent 92% of ISOPO$_2$ loss, with the remainder due to reactions with organic peroxy radicals.

Under high-$NO_x$ conditions, CHOCHO is produced promptly via products of the $\delta$ isomers (HC5, DIBOO) (Paulot et al.,

2009a; Galloway et al., 2011). CHOCHO production via the $\beta$ isomers is slower, due to the intermediary production of methylvinylketone (MVK) followed by glycolaldehyde (GLYC). GEOS-Chem originally had a fixed $\delta$ branching ratio of 24% for the reaction of ISOPO$_2$ + NO, based on the chamber experiments of Paulot et al. (2009a). However recent work has shown that $O_2$ addition to the isoprene-OH adducts is reversible (pink pathway, Figure 1), allowing interconversion between $\beta$ and $\delta$ ISOPO$_2$ isomers (Peeters et al., 2009; Crounse et al., 2011; Peeters et al., 2014). $\beta$ isomers are heavily favoured at

equilibrium, accounting for $\sim 95\%$ of ISOPO$_2$ (Peeters et al., 2014). The experimental conditions in Paulot et al. (2009a) used high NO concentrations ($\sim 500$ ppbv). This implies short ISOPO$_2$ lifetimes, and thus may not reflect the degree of isomer interconversion seen at ambient oxidant levels. Here we adopt a $\delta-$ISOPO$_2$ branch ratio of 10%, following Fisher et al. (2016), to match SEAC$^4$RS observations of organic nitrates produced through the $\delta-$ISOPO$_2$ + NO pathway.

CHOCHO forms under low-$NO_x$ conditions through isoprene epoxydiols (IEPOX) and through the ISOPO$_2$ isomerization

pathway. IEPOX forms as second-generation non-radical product of isoprene oxidation via ISOPOOH, and thus represents a slow CHOCHO formation pathway. IEPOX isomer fractions in GEOS-Chem are based on equilibrium $\delta/\beta$ ISOPO$_2$ branching ratios (Bates et al., 2014; Travis et al., 2016). At low $NO_x$ levels the ISOPO$_2$ lifetime is sufficiently long for equilibrium to be reached (Peeters et al., 2014). ISOPO$_2$ isomerization in the previous GEOS-Chem mechanism of Travis et al. (2016) produced solely hydroperoxyaldehydes (HPALDs), but here we also include the formation of dihydroperoxy $\alpha$-formyl peroxy radicals

(di-HPCARPs) (Peeters et al., 2014) following the Master Chemical Mechanism v3.3.1 (MCMv3.3.1) (Jenkin et al., 2015). di-HPCARPs in MCMv3.3.1 have a low CHOCHO yield, but here we introduce a (1,5)H-shift isomerization of di-HPCARPs that could be competitive with the (1,4)H-shift isomerization due to the presence of the terminal-peroxide functional group (Crounse et al., 2013). The resulting di-hydroperoxide dicarbonyl compound (DHDC) product quickly photolyzes to produce



CHOCHO, analagous to the mechanisms proposed for HPALDs (Peeters et al., 2014) and carbonyl nitrates (Müller et al., 2014). We find that this pathway can explain SENEX observations of prompt CHOCHO production under low-$NO_x$ conditions.

GEOS-Chem includes CHOCHO loss via photolysis and oxidation by OH. Pressure-dependent CHOCHO photolysis rates are computed using the FAST-JX radiative transfer model (http://www.ess.uci.edu/~prather/fastJX.html). CHOCHO loss via aerosol reactive uptake does not significantly alter daytime CHOCHO concentrations because the CHOCHO lifetime against OH and photolysis is short (1-2 h). Since we only consider daytime observations (10-17 local), our model evaluation is not sensitive to aerosol reactive uptake, in contrast with a previous CTM comparison to SENEX by Li et al. (2016) where no time filtering was applied.

### 2.3 Time- and $NO_x$-dependent CHOCHO and HCHO yields from isoprene

Understanding the time- and $NO_x$-dependent yields of CHOCHO and HCHO from isoprene oxidation is critical for interpreting observed CHOCHO and HCHO columns from space in terms of isoprene emissions. Here we examine time-dependent CHOCHO and HCHO molar yields in the GEOS-Chem and MCMv3.3.1 chemical mechanisms using the DSMACC box model (Emmerson and Evans, 2009). Simulations are initiated at 9am local time with 1 ppbv isoprene, 40 ppbv $O_3$, and 100 ppbv CO. $NO_x$ concentrations are held at fixed values. Photolysis rates are calculated for clear-sky with the TUV radiative transfer model (Madronich, 1987). To correct for differences in time-dependent yields associated with differences in OH concentrations, we reference GEOS-Chem and MCMv3.3.1 results to a common "OH exposure time" variable ($t_{OH}$);

$$t_{OH} = \frac{1}{[OH]_{ref}} \int_0^t [OH](t')dt' \qquad (1)$$

Here $[OH](t)$ is the OH concentration simulated in the box model, and $[OH]_{ref} = 4 \times 10^6$ molecules $cm^{-3}$ is a reference OH concentration representative of summer daytime conditions over the Southeast US (Wolfe et al., 2016). For a fixed $[OH] = 4 \times 10^6$ molecules $cm^{-3}$ $t_{OH}$ represents the actual time.

Figure 2 shows the time- and $NO_x$-dependent cumulative molar yields of CHOCHO and HCHO in GEOS-Chem and MCMv3.3.1. The branching ratio of $ISOPO_2$ as a function of $NO_x$ is also shown. The time-dependent HCHO yields in both mechanisms are similar under high-$NO_x$ conditions. Additional confidence in the HCHO yield under these conditions is offered by the ability of GEOS-Chem to reproduce the observed correlation between HCHO and isoprene organic nitrates (Mao et al., 2013; Fisher et al., 2016). The HCHO yield is lower under low-$NO_x$ conditions in both mechanisms, and overall the difference between them is minor.

There is far more disagreement between the two mechanisms for CHOCHO yields. Under high-$NO_x$ conditions, GEOS-Chem produces CHOCHO rapidly in the first two hours due to its higher $\delta-ISOPO_2 + NO$ branching ratio (10% in GEOS-Chem vs. 3.4% in MCMv3.3.1). This is compensated at longer OH-exposure times by higher GLYC yields from isoprene in MCMv3.3.1. GEOS-Chem produces higher ultimate yields of CHOCHO under low-$NO_x$ conditions mainly due to DHDC formation and subsequent photolysis, neither of which are included in MCMv3.3.1. The $NO_x$-dependence of the CHOCHO





yield in MCMv3.3.1 is similar to that of HCHO, implying that CHOCHO and HCHO observations would provide redundant information on isoprene emissions. The SENEX observations indicate that CHOCHO yields under low-$NO_x$ conditions are too low in MCMv3.3.1, as discussed below. In GEOS-Chem, by contrast, the CHOCHO and HCHO yields show opposite dependences on $NO_x$, implying that they could provide complementary information on isoprene emissions.

## 3   Constraints from SENEX observations

Figure 3 shows the observed and simulated median vertical profiles of CHOCHO, HCHO, and $NO_x$ concentrations along the SENEX flight tracks. Figure 4 shows maps of concentrations below 1 km altitude (above ground level) taken as the mixed layer. Here and elsewhere we only include daytime observations (10-17 local) and exclude targeted sampling of biomass burning plumes (diagnosed by acetonitrile concentrations above 200 pptv). CHOCHO, HCHO and $NO_x$ were measured by the Airborne Cavity Enhanced Spectrometer (ACES) (Min et al., 2016), In-Situ Airborne Formaldehyde (ISAF) instrument (Cazorla et al., 2015), and the NOAA chemiluminescence instrument (Ryerson et al., 1999; Pollack et al., 2010), with stated accuracies of 6%, 10%, and 5% respectively.

Simulated median $NO_x$ concentrations in the mixed layer are within 10% of observations, supporting the 50% reduction in EPA NEI $NO_x$ emissions previously inferred from the analysis of SEAC[4]RS observations by Travis et al. (2016), also included here (Section 2.1). Half of isoprene oxidation in the model under the SENEX conditions takes place by the low-$NO_x$ pathways (Figure 1). Simulated median CHOCHO and HCHO concentrations in the mixed layer are within 20% of observations, but the model is too low at higher altitudes. During SENEX the mixed layer was typically capped by a neutrally stable transition layer of shallow cumulus convection which extended up to 3 km (Wagner et al., 2015), suggesting that transport via this mechanism is underestimated in the model. The CHOCHO observations in the free troposphere ($> 3$ km) have to be treated with caution since they are close to the instrument detection limit (Kaiser et al., 2015). It is therefore difficult to determine whether the bias is due to a missing CHOCHO source in the model or instrument artifact. Elevated CHOCHO concentrations above the boundary layer have also been observed in previous campaigns over the Southeast US (Lee et al., 1998), California (Baidar et al., 2013), and the remote Pacific (Volkamer et al., 2015). There could be a free tropospheric source missing in the model, but it is unclear what this source could be, and correlative analysis in the SENEX observations offer no insight.

The mixed layer concentrations maps in Figure 4 show that the model captures some of the horizontal variability in the observations. The spatial correlation for HCHO is high ($r = 0.75$) as in SEAC[4]RS ($r = 0.64$, Zhu et al. (2016)), and reflects isoprene emission patterns. Correlation for CHOCHO is also relatively strong ($r = 0.51$). Average mixed layer CHOCHO and HCHO concentrations simulated by the model for the SENEX period are much more uniform than those sampled along the SENEX flight tracks, as shown in the GEOS-Chem panels of Figure 4. This is because of day-to-day variability in isoprene emissions, mostly driven by temperature (Zhu et al., 2016).

Figure 5 compares simulated and observed CHOCHO vs. HCHO relationships in the mixed layer color coded by $NO_x$ concentrations. Correlation between the two species is strong, and model and observations are consistent. This might suggest



that CHOCHO and HCHO provide redundant information for constraining isoprene emissions. However, examination of Figure 5 indicates some $NO_x$ sensitivity, which will be discussed further below.

Measurements of isoprene (ISOP) and total methylvinylketone + methacrolein (MVK+MACR) made by proton transfer mass spectrometry from the SENEX aircraft (de Gouw and Warneke, 2007) allow some evaluation of GEOS-Chem CHOCHO and HCHO yields by using the parcel model of Wolfe et al. (2016) to infer initial isoprene $[ISOP]_0$ and OH exposure time $t_{OH}$. The evolution of ISOP and MACR concentrations within the parcel is given by

$$ISOP + OH \xrightarrow{k_1} Y_{MACR}(NO)MACR \tag{R1}$$

$$MACR + OH \xrightarrow{k_2} products \tag{R2}$$

Here $Y_{MACR}(NO)$ is the NO-dependent yield of MACR from isoprene, and $k_1$ and $k_2$ are the rate constants for the reactions of OH with ISOP and MACR respectively, all given by Wolfe et al. (2016). Whole air samples (Lerner et al., 2016) during SENEX indicated a uniform $[MVK]/[MACR]$ ratio of $2.3 \pm 0.2$ mol mol$^{-1}$ so that the MACR concentrations can be inferred from the higher-frequency MVK+MACR measurement. For an air parcel initially containing only isoprene we derive the following expression for the ratio of MACR to isoprene as sampled by the aircraft.

$$\frac{[MACR]}{[ISOP]} = \frac{Y_{MACR}(NO)k_1}{k_2 - k_1}\left(1 - \exp\left((k_1 - k_2)[OH]_{ref}t_{OH}\right)\right) \tag{2}$$

We use equation 2 to calculate $t_{OH}$ from the observed $[MACR]/[ISOP]$ ratios and from there to infer the initial isoprene concentration $[ISOP]_0=[ISOP]\exp(k_1[OH]_{ref}t_{OH})$. The calculation is applied to the ensemble of SENEX data below 1 km altitude and yields $t_{OH}$ in the range of 0.25 - 1.5 h.

Figure 6 shows the observed relationships of CHOCHO and HCHO concentrations vs. initial isoprene color coded by $NO_x$ concentrations. High-$NO_x$ and low-$NO_x$ conditions can be separated by envelopes using linear regression fits to the data with $NO_x$ concentrations above 800 pptv and below 200 pptv respectively (Figure 2). The observed slopes increase by at least a factor of two for both CHOCHO and HCHO in the transition from low- to high-$NO_x$ conditions. This is well reproduced by GEOS-Chem, even though the model CHOCHO yield is $NO_x$-independent over the first few hours of isoprene oxidation (Figure 2). The higher CHOCHO under high-$NO_x$ conditions in GEOS-Chem is due to longer photochemical aging, as OH concentrations increase with increasing $NO_x$. Overall the comparison in Figure 6 provides support for the CHOCHO and HCHO yields computed by GEOS-Chem and their dependences on $NO_x$.

Previous studies have used the $R_{GF}=[CHOCHO]/[HCHO]$ ratio as an indicator for different VOC precursors (Vrekoussis et al., 2010; DiGangi et al., 2012). In the Southeast US, isoprene is the dominant source of both. In this case variations in $R_{GF}$ would be expected to reflect differences in the chemical environment for isoprene oxidation, and the information may be useful for relating satellite column observations to isoprene emission. Figure 7 shows $R_{GF}$ as a function of $NO_x$ below 1 km





in the SENEX observations and as simulated by GEOS-Chem. Points are color coded by OH exposure time $t_{OH}$ (Equation 1), derived from the parcel model. The median and interquartile $R_{GF}$ values binned in 250 pptv $NO_x$ increments are also shown. The observed median $R_{GF}$ values (0.02 to 0.024 mol mol$^{-1}$) show no significant dependence on $NO_x$, while GEOS-Chem shows a weak dependence. In both the model and observations there is a subset of low-$NO_x$ points with higher $R_{GF}$ values

(0.03-0.06). These correspond to short OH exposure times and are caused by OH titration by isoprene. The high $R_{GF}$ reflects the relatively faster production of CHOCHO than HCHO in the early stage of isoprene oxidation under low-$NO_x$ conditions as shown by Figure 2. The presence of that population in the observations provides support for fast glyoxal production from the isomerization pathway of isoprene oxidation (Figure 1) that is present in GEOS-Chem but not in MCMv3.3.1.

## 4  Implications for satellite observations

Knowledge gained from SENEX enables an improved interpretation of CHOCHO and HCHO column observations from space in isoprene dominated environments. We use for this purpose June-August 2006-2007 observations of CHOCHO, HCHO, and tropospheric $NO_2$ columns from the Ozone Monitoring Instrument (OMI). OMI was launched onboard the NASA Aura satellite in July 2004, and provides daily global coverage in sun-synchronous orbit with an equatorial crossing time of 13:40 LT. The CHOCHO data are from the Smithsonian Astrophysical Observatory (SAO) retrieval described in Chan Miller et al. (2014) and

hereby referred to as OMI SAO. The HCHO and $NO_2$ data are from the OMI Version 3 product release (González Abad et al., 2015; Bucsela et al., 2013). Retrievals are in the 435-461 nm spectral range for CHOCHO, 328.5-356.5 nm for HCHO, and 405-465 nm for $NO_2$. We use 2006-2007 data because 2013 data for CHOCHO are very noisy (Figure S1), possibly because of sensor degradation.

Slant columns along the optical path of the backscattered solar radiation are fitted to the observed spectra and converted to
vertical columns by division with an air mass factor (AMF) that accounts for the viewing geometry, atmospheric scattering, and the vertical profile of the gas (Palmer et al., 2001):

$$AMF = \int_0^\infty w(z)s(z)dz \qquad (3)$$

Here $w(z)$ is the scattering weight measuring the sensitivity of the retrieval to the gas concentration at altitude $z$, and $s(z)$ is a normalized vertical profile of gas number density. Here we recomputed the AMFs for the three retrievals using vertical
profiles from GEOS-Chem, as it is necessary for comparing simulated and observed vertical columns (Duncan et al., 2014). We remove observations impacted by the row anomaly (http://www.knmi.nl/omi/research/product/rowanomaly-background.php), and those with cloud fractions over 20%. Previous validation of the OMI HCHO retrievals with SEAC$^4$RS aircraft observations revealed a 43% uniform low bias (Zhu et al., 2016), corrected in the data shown here.

Figure 8 compares CHOCHO and HCHO vertical columns from GEOS-Chem and OMI, and Figure 9 shows spatial cor-
relations over the eastern US. Excellent agreement is found for HCHO, providing an independent test of the correction to the OMI HCHO retrieval inferred from the SEAC$^4$RS data (Zhu et al., 2016). CHOCHO from OMI is highly correlated with



GEOS-Chem ($r = 0.76$), indicative of the isoprene source. However OMI CHOCHO shows a higher continental background and a factor of 2 weaker enhancement over the Southeast US.

Zhu et al. (2016) suggested that errors in the assumed surface reflectivities affecting the AMFs were an important source of the bias in the OMI HCHO retrievals. CHOCHO retrievals are even more sensitive to surface reflectivity because of the longer wavelengths. Russell et al. (2011) previously pointed out that the OMI surface reflectivities used in the standard $NO_2$ retrievals (Kleipool et al., 2008) were too high and replaced them with high resolution ($0.05° \times 0.05°$) reflectivity observations from MODIS (Schaaf and Wang, 2015) to produce the Berkeley High-Resolution (BEHR) OMI $NO_2$ retrieval. CHOCHO and $NO_2$ are retrieved at similar wavelengths so the sensitivity to surface reflectivity should be similar. Figure 8 (bottom right) shows the mean CHOCHO scattering weights computed from the OMI-SAO and BEHR. The lower BEHR surface reflectivity values result in a lower AMF and hence a higher vertical column (Figure 8, bottom left panel). The slope of the regression between GEOS-Chem and OMI CHOCHO columns increases from 0.46 to 0.57, improving but not reconciling the differences.

As pointed out above, SENEX and other observations suggest that GEOS-Chem may be missing a source of CHOCHO in the free troposphere (Figure 3), although it is not clear what this source might be. Integration of the median CHOCHO profile above 2 km in Figure 3 shows a negative model bias of $1.3 \times 10^{14}$ molecules cm$^{-2}$, comparable to the continental background intercept in Figure 9 ($1.7 \times 10^{14}$ molecules cm$^{-2}$). The presence of free tropospheric CHOCHO would further impact the AMF calculation under continental background conditions since the retrieval sensitivity as measured by the scattering weights increases with altitude. Thus the retrieved continental background would be overestimated.

Figure 10 shows CHOCHO vs. HCHO relationships for OMI (using the BEHR scattering weights) and GEOS-Chem, color coded by tropospheric $NO_2$ columns. Individual points are seasonal averages (data points from Figure 8) in order to limit noise. The slope is steeper in GEOS-Chem because the CHOCHO columns are higher. Since GEOS-Chem reproduces the aircraft CHOCHO-HCHO relationship without bias (Figure 5), this is further evidence of bias in the OMI CHOCHO observations. The CHOCHO-HCHO relationship is tight in both OMI ($r = 0.83$) and GEOS-Chem ($r = 0.99$), with no indication of a separate population of low-$NO_x$ points with high $R_{GF}$ as there was in the SENEX data. It thus appears from the OMI data that satellite observations of CHOCHO and HCHO in isoprene-dominated environments are redundant. This may reflect the higher $NO_x$ levels in 2006-2007 compared to 2013 (Russell et al., 2012). However since median $R_{GF}$ shows no significant variation with $NO_x$ in the SENEX data (Figure 7), the required temporal averaging of satellite observations is a more likely explanation for the tight correlation. Finer-scale and more temporally resolved data, as will be available from the TEMPO geostationary instrument to be launched in the 2018-2020 time frame (Zoogman et al., 2016), may provide new perspectives of the utility of the CHOCHO retrieval.

# 5 Conclusions

We have used aircraft observations of glyoxal (CHOCHO), formaldehyde (HCHO), and related species from the SENEX aircraft campaign over the Southeast US together with OMI satellite data to better understand the CHOCHO yield from isoprene



and the complementarity of CHOCHO and HCHO observations from space for constraining isoprene emissions. This work includes a first validation of the CHOCHO retrieval from the OMI satellite instrument.

We began with an analysis of the time- and $NO_x$ dependent CHOCHO and HCHO yields from isoprene oxidation in the GEOS-Chem chemical transport model and in the Master Chemical Mechanism (MCMv3.3.1). The GEOS-Chem mechanism features several updates relevant to CHOCHO formation. These include a decrease in the $\delta-ISOPO_2 + NO$ branching ratio leading to prompt CHOCHO production under high-$NO_x$ conditions, and a low-$NO_x$ pathway for prompt CHOCHO formation from a (1,5)H-shift isomerization of dihydroperoxy $\alpha$-formyl peroxy radicals formed through the $ISOPO_2$ isomerization pathway (proposed here). GEOS-Chem and MCMv3.3.1 show similar HCHO yields from isoprene, increasing with increasing $NO_x$. CHOCHO yields from isoprene in MCMv3.3.1 show behavior similar to HCHO but GEOS-Chem has a higher yield at low-$NO_x$ from the $ISOPO_2$ isomerization pathway.

Comparison of GEOS-Chem to the SENEX observations of CHOCHO and HCHO shows good agreement in the boundary layer but a negative CHOCHO model bias in the free troposphere. This could reflect an instrument artifact but may also imply a missing background source in the model. Mixed layer ($< 1$ km) observations show a strong CHOCHO-HCHO relationship that is reproduced in GEOS-Chem and is remarkably consistent across all conditions except at very low $NO_x$ where the [CHOCHO]/[HCHO] ratio ($R_{GF}$) can be unusually high. This reflects prompt formation of CHOCHO under low-$NO_x$ conditions, which the model attributes to the new pathway via $ISOPO_2$ isomerization followed by DHDC photolysis proposed here (Figure 1).

The SENEX observations enable indirect validation of the OMI CHOCHO satellite data using GEOS-Chem as an inter-comparison platform. The OMI data show a continental background that is consistent with the SENEX free tropospheric observations, and an enhancement over the Southeast US that is consistent with the isoprene source. However this enhancement is a factor of 2 too low in the OMI data. A partial explanation is that surface reflectivities assumed in the retrieval are too high. The satellite data show strong CHOCHO-HCHO correlation consistent with the model and imply that the two gases provide redundant information for constraining isoprene emissions in regions where isoprene is their dominant precursor. This may reflect the seasonal averaging in the OMI data required to reduce noise, which still may permit observation of oxidation pathway-driven changes in the CHOCHO-HCHO relationship from future geostationary missions.

*Acknowledgements.* This work was funded by NASA ACMAP and ACCDAM and is a contribution to the NASA Aura Science Team. This research was undertaken with the assistance of resources provided at the NCI National Facility systems at the Australian National University through the National Computational Merit Allocation Scheme supported by the Australian Government.



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





**Figures**



**Figure 1.** Pathways for glyoxal (CHOCHO) formation from isoprene oxidation in GEOS-Chem as implemented in this work. Only species relevant to CHOCHO formation are shown. Branching ratios, species lifetimes, and contributions to glyoxal and glycolaldehyde (GLYC) formation from each boxed species are mean values over the Southeast United States ( $96.25 - 73.75°$W, $29 - 41°$N ) during the SENEX campaign (June 1st - July 10th 2013). Species lifetimes are shown for an OH concentration of $4 \times 10^6$ molecules cm$^{-3}$.





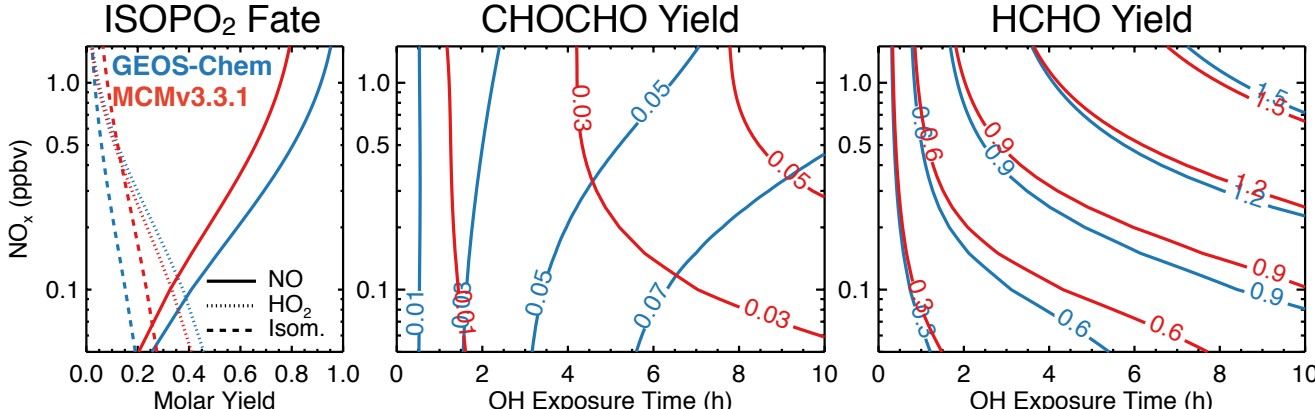

**Figure 2.** Cumulative time- and $NO_x$ dependent molar yields of CHOCHO and HCHO from isoprene oxidation in the GEOS-Chem and MCM3.3.1 chemical mechanisms. Results are from box model simulations with fixed $NO_x$ concentration as described in the text, and are presented as functions of the imposed $NO_x$ concentration (vertical axis). The left panel shows the isoprene peroxy radical ($ISOPO_2$) branching ratios for reaction with NO, $HO_2$, and isomerization. The middle and right panels show the time-dependent cumulative yields of CHOCHO and HCHO, where time is normalized by OH exposure (Equation 1). "OH exposure time" is equivalent to time for a constant $[OH] = 4 \times 10^6$ molecules $cm^{-3}$.

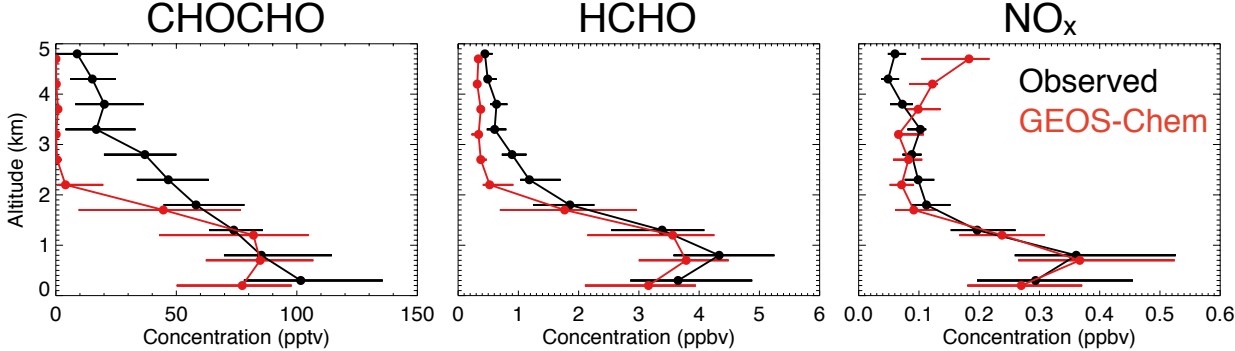

**Figure 3.** Median vertical profiles of CHOCHO, HCHO, and $NO_x$ concentrations during SENEX (June 1 - July 10 2013). Observed concentrations (Min et al., 2016; Cazorla et al., 2015; Pollack et al., 2010) are compared to GEOS-Chem model values sampled along the flight tracks. Horizontal bars indicate interquartile range. Altitudes are above ground level (AGL).





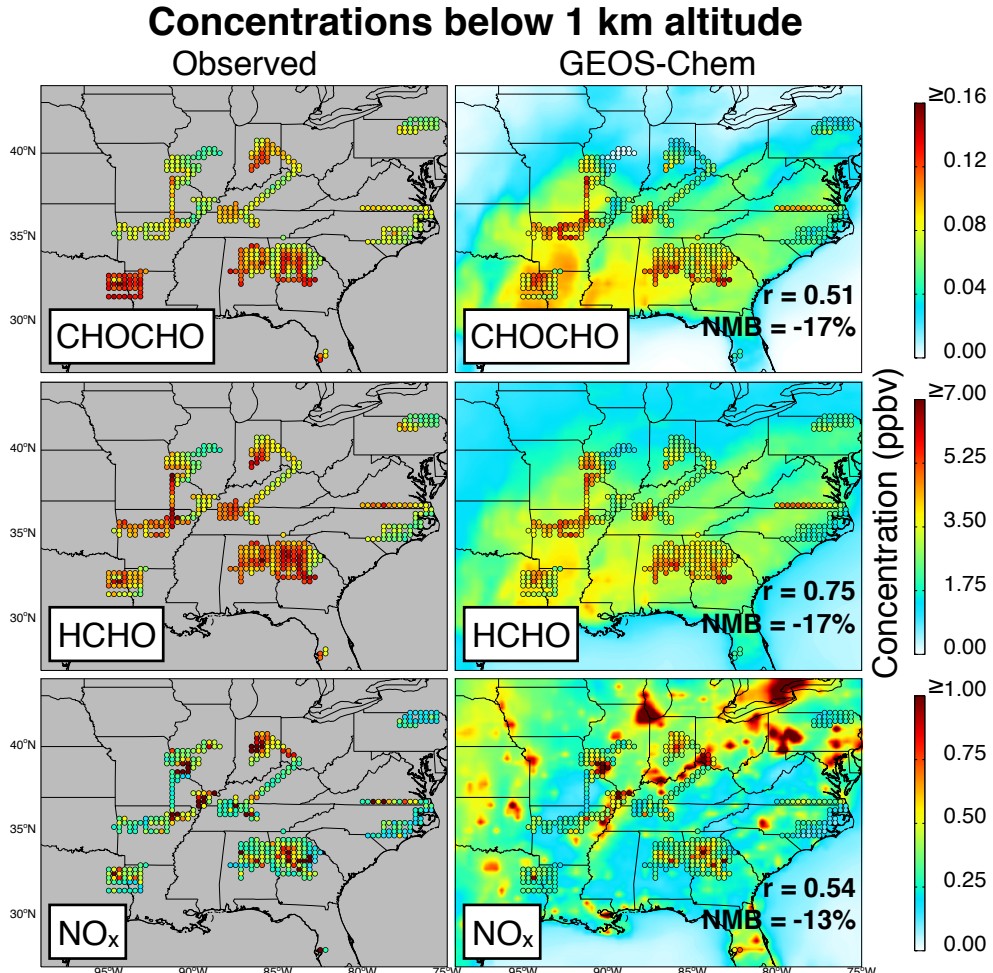

**Figure 4.** CHOCHO, HCHO, and $NO_x$ concentrations below 1 km AGL during SENEX (June 1 - July 10 2013). The grid squares show daytime aircraft observations compared to the colocated GEOS-Chem model values on the $0.25° \times 0.3125°$ model grid. Background contours in the right panels show the average model-simulated concentrations at 13 - 14 local time for the SENEX period. Comparison statistics between model and observation grid squares are shown as the correlation coefficient $r$ and the normalized mean bias (NMB). Correlation statistics for $NO_2$ exclude urban plumes in the observations ($[NO_x] > 4$ ppb) as these would not be resolved at the scale of the model.



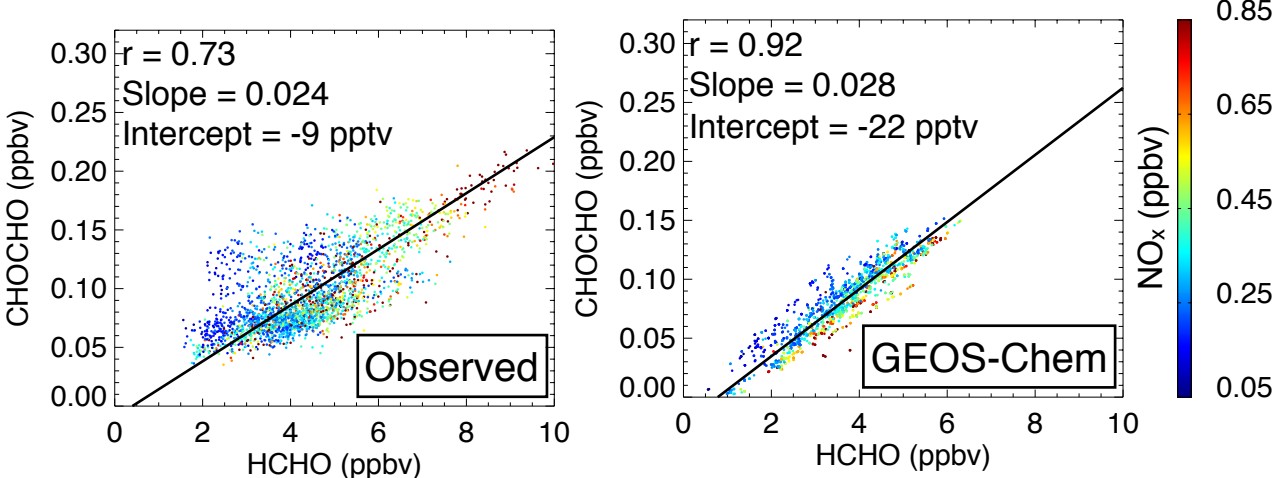

**Figure 5.** Relationship between CHOCHO and HCHO concentrations in the mixed layer (< 1 km AGL) during SENEX (June 1 - July 10 2013), color coded by $NO_x$ concentration. Observed concentrations (Min et al., 2016; Cazorla et al., 2015) are compared to GEOS-Chem model values sampled along the flight tracks. Lines and reported slopes are from reduced major axis regressions.





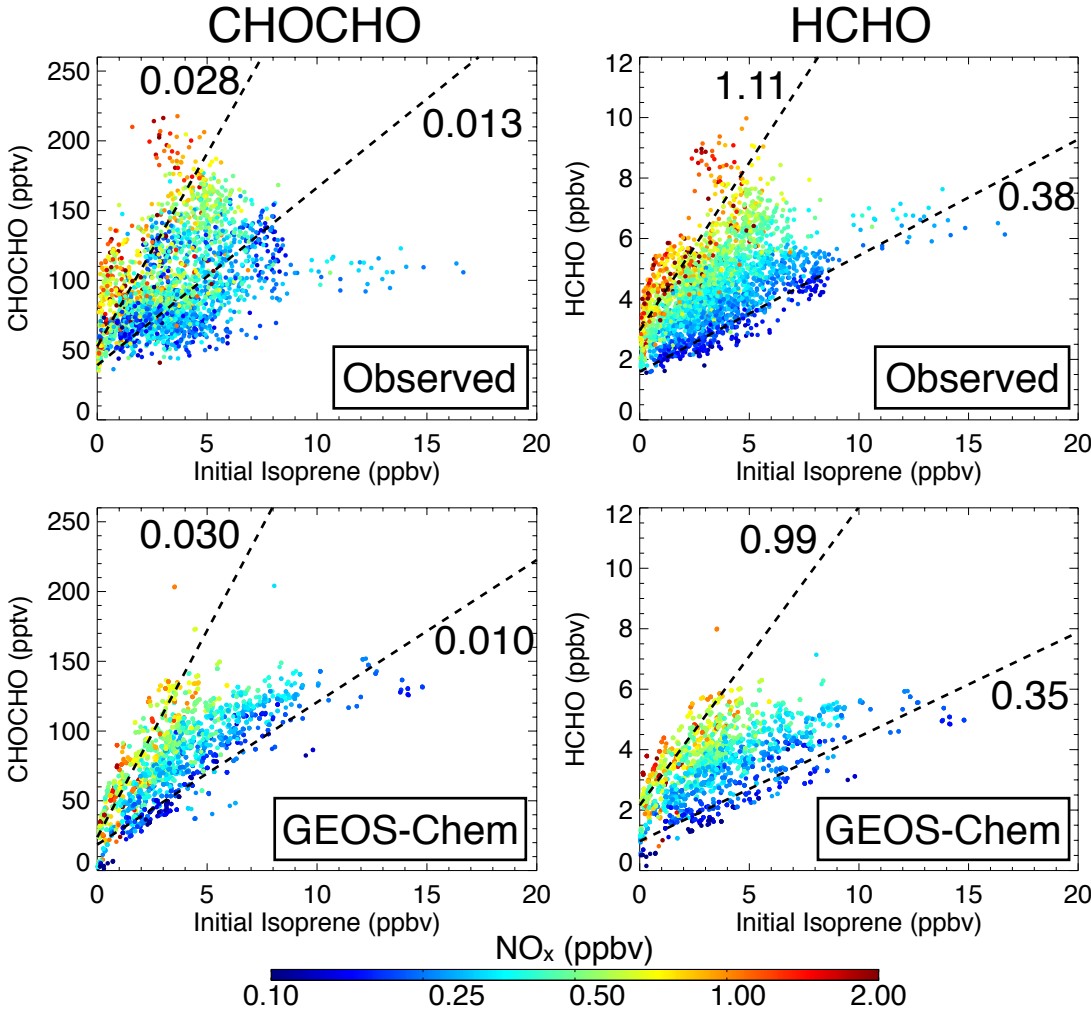

**Figure 6.** Relationships of mixed layer CHOCHO and HCHO concentrations in SENEX to initial isoprene as derived from a parcel model. Observations are compared to the corresponding GEOS-Chem model values sampled along the aircraft flight tracks. Points are color coded by $NO_x$ concentration. Dashed lines are reduced major axis regression fits to data below 200 pptv and above 800 pptv $NO_x$. The slopes from the regressions are indicated next to the dashed lines.



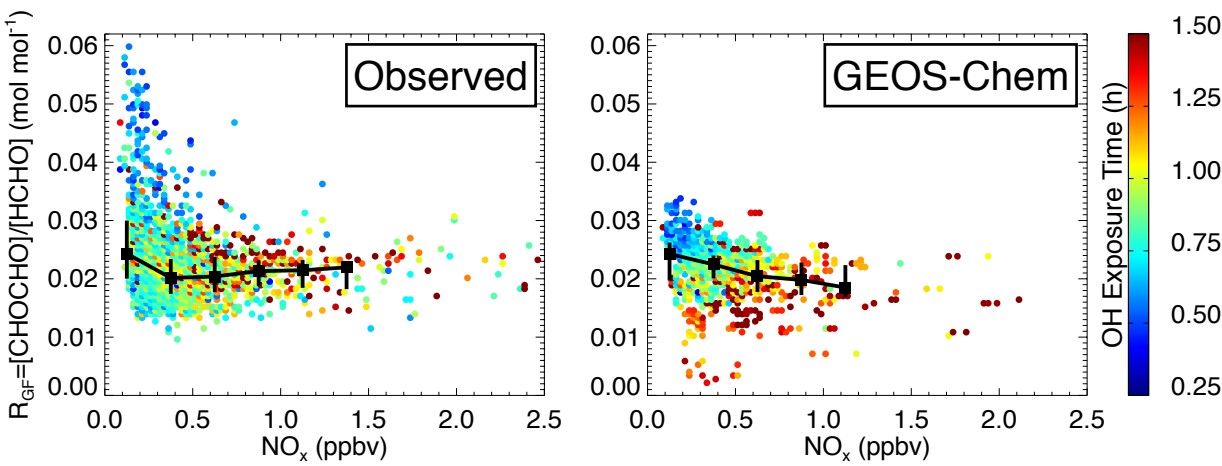

**Figure 7.** Dependence of the CHOCHO-to-HCHO ratio $R_{GF}$ on $NO_x$ concentrations for the SENEX conditions. Observations below 1 km altitude (left) are compared to GEOS-Chem model values sampled along the flight tracks (right). Points are color coded by the OH exposure time $t_{OH}$ (Equation 1). Binned median and interquartile $R_{GF}$ values in increments of 250 pptv $NO_x$ for bins with more than 20 values are also shown.





**Figure 8.** Mean CHOCHO and HCHO columns in summer (JJA) 2006-2007. GEOS-Chem model values (top) are compared to OMI satellite observations (middle and bottom). OMI-SAO is the standard operational product (Chan Miller et al., 2014; González Abad et al., 2015). The OMI-BEHR product for CHOCHO uses tropospheric scattering weights from the BEHR $NO_2$ retrieval (Russell et al., 2011; Laughner et al., 2016). The OMI HCHO observations have been scaled up by a factor of 1.67 to correct for retrieval bias (Zhu et al., 2016). The normalized mean bias ($NMB$) between GEOS-Chem and OMI in the Southeast US ($75° - 100°W, 29.5° - 37.5°N$) is shown within the GEOS-Chem panels. The bottom right panel shows the mean CHOCHO scattering weights ($w$) from the OMI-SAO and OMI-BEHR retrievals and the vertical shape factors ($s$) over the Southeast US from the SENEX observations and GEOS-Chem in the Southeast US from a typical orbit (10114, 9 June 2006).




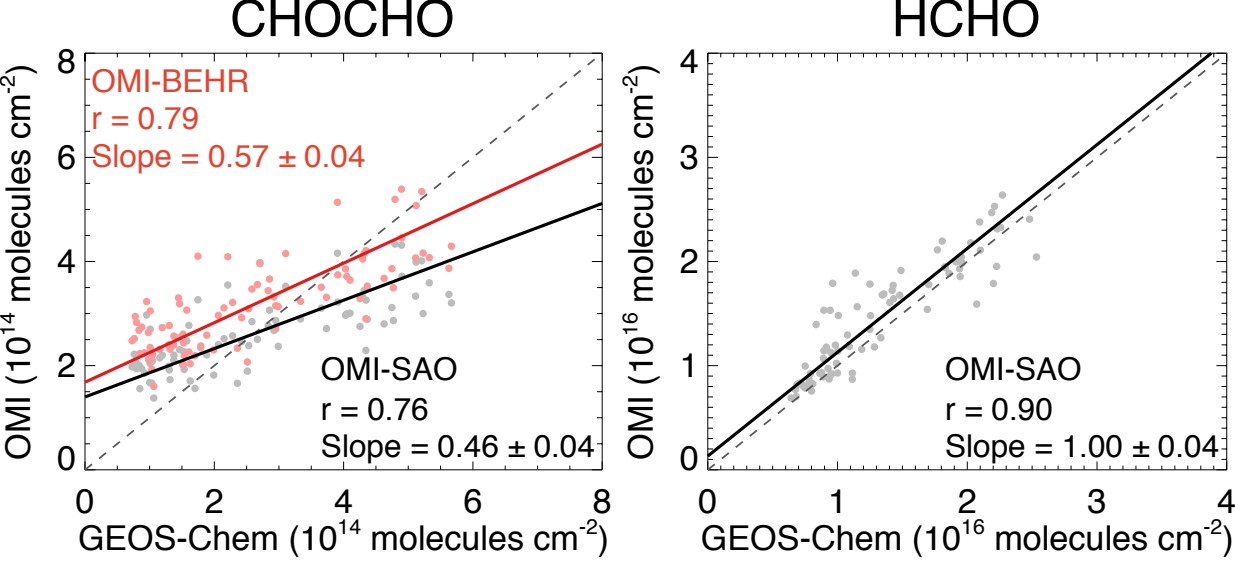

**Figure 9.** scatter plots of OMI vs. GEOS-Chem CHOCHO and HCHO columns over the eastern US ($75° - 100°$W, $29.5° - 45°$N). Values are seasonal means for JJA 2006-2007 as plotted in Figure 8. OMI observations for CHOCHO are from the standard SAO retrieval (Chan Miller et al., 2014) and using BEHR scattering weights (Russell et al., 2011; Laughner et al., 2016). Correlation coefficients and reduced-major-axis (RMA) regressions are shown.

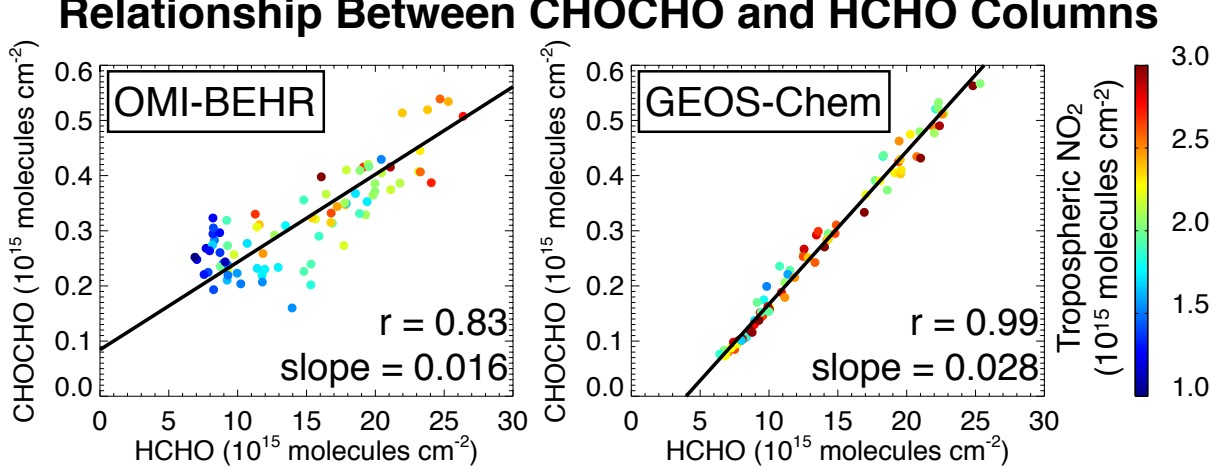

**Figure 10.** Relationship between CHOCHO and HCHO vertical columns over the eastern US ($75° - 100°$W, $29.5° - 45°$N) in June-August 2006-2007 color coded by tropospheric $NO_2$ columns. OMI values with CHOCHO AMFs computed from BEHR scattering weights are compared to GEOS-Chem values. Lines and reported slopes are from reduced major axis regressions.