# Peer review of "Glyoxal yield from isoprene oxidation and relation to formaldehyde: chemical mechanism, constraints from SENEX aircraft observations, and interpretation of OMI satellite data"

_Atmospheric Chemistry and Physics, 2016_

## Short Comment (SC1) · 30 Nov 2016

We recently published a paper on glyoxal production from isoprene oxidation using SENEX observations (Li et al., 2016). It might be useful for authors to discuss in some details about the commonality and differences between Li et al. (2016) and this work.

Reference

Li, J., Mao, J., Min, K.-E., Washenfelder, R. A., Brown, S. S., Kaiser, J., Keutsch, F. N., Volkamer, R., Wolfe, G. M., Hanisco, T. F., Pollack, I. B., Ryerson, T. B., Graus, M., Gilman, J. B., Lerner, B. M., Warneke, C., de Gouw, J. A., Middlebrook, A. M., Liao, J.,

[Figure]

Welti, A., Henderson, B. H., McNeill, V. F., Hall, S. R., Ullmann, K., Donner, L. J., Paulot, F., and Horowitz, L. W.: Observational constraints on glyoxal production from isoprene oxidation and its contribution to organic aerosol over the Southeast United States, Journal of Geophysical Research: Atmospheres, n/a-n/a, 10.1002/2016JD025331, 2016.

---

## Referee Comment (RC1) · Anonymous Referee #1 · 16 Dec 2016

General Comments:

The authors present a studying using GEOS-Chem and box model simulations to understand and interpret observations of glyoxal and formaldehyde from the SENEX aircraft campaign over the Eastern US. In addition, they compare results to satellite-derived formaldehyde and glyoxal columns to determine if there is separate information about isoprene emissions that can be obtained from each species (which they do not find). The manuscript is well written and should be published after addressing the following minor comments.

Specific Comments:

1. Beta vs delta isoprene RO2 isomers: Can you clarify the yield of beta vs delta RIO2 isomers separately from the RIO2+NO ISOPN yield from each isomer? Table S1 indicates the authors recommend an update to the RIO2+NO → ISOPNB and ISOPND yields. I wonder how much of that update is due to the different isomer distribution and how much is due to the yield of ISOPN from each isomer. Note that in older versions of the isoprene chemistry, the yield of beta-RIO2+NO to produce ISOPNB and delta-RIO2+NO to produce ISOPND were different. Fisher et al., 2016 updated them to both be 9%. The 10% yield of delta isomers indicated in Figure 1 is higher than MCM (3.4%, page 5 line 29).

2. The Li et al. paper is only cited twice despite using a very similar data set and working on a similar issue. More synthesis of results in the context of Li et al. would be helpful. For example, do both models agree in terms of the role of RO2 isomerization and its contribution to glyoxal? Comparing Figure 1 of Li et al. to Figure 1 in this paper indicates Li et al. predict a much larger role for RO2+HO2 relative to isomerization in producing glyoxal (but the figures are not directly comparable, so it is not clear).

3. Page 5, line 6-7. How was it determined that the model was not sensitive to aerosol reactive uptake? Was that through a simulation or estimated lifetime against uptake? The authors note that a background/free tropospheric source of glyoxal may be missing from the model. Have the authors considered whether or not reversible uptake of glyoxal, particularly if it is formed in the boundary layer and repartitions to glyoxal in the free troposphere, may provide this missing source?

4. Page 8, near line 5 and Figure 7: The model shows a population of points with R(GF) < 0.01 while observations do not indicate such low R(GF) at any time. Do you know the cause of these low modeled R(GF)?

5. Page 9, line 26-27 indicates finer scale, more temporally resolved data may provide valuable glyoxal data from satellite? Are the authors hypothesizing that R(GF)s may

be more variable? Can GEOS-Chem predictions be used to test that theory?

6. Figure 6: Is the influence of NOx due to the effect on RO2 branching or OH?

Technical Corrections:

7. Second sentence of abstract could be reworded as it is not clear if HCHO is also measureable from space via the same technique as glyoxal or not.

8. Page 3, line 12: "is in better agreement" than Vrekoussis?

9. Page 4, line 15: delta "vs beta" branching ratio

10. Page 4, line 15: forms as "a" second-generation

---

## Referee Comment (RC2) · Anonymous Referee #2 · 29 Dec 2016

The analyses of CHOCHO and HCHO in this paper have many interesting components, three models (GEOS-Chem, DSMACC, and a parcel model), two mechanisms (GEOS-Chem and MCM), SENEX in situ observations, and OMI retrievals. A casual reading would suggest it is a publishable paper. But in the more careful second-round reading, I found many problems. I cannot recommend publishing this paper in its present form. Substantial changes are required.

This work was probably done during the same period as Li et al. (2016, Observational constraints on glyoxal production...). The publication of that paper makes it necessary

that differences between the two papers are resolved in this paper. Very little was done in this paper. Many differences were not mentioned. For example, Li et al. (2016) showed some effects of aerosol loss of CHOCHO in the mixed layer. Their budget shows that aerosol loss is 26% of total CHOCHO loss in the boundary layer, which is quite significant. The justification for not including this loss given in line 4-8 on P. 5 did not provide either the details on aerosol loss modeling or the results.

Many analyses in this paper are similar to Li et al. (2016), but the results are different. The implications of these differences were not considered in this paper. The omission of comparing the simulation of isoprene to the observations, which was done by Li et al., may be an indication that the submission of this paper was rushed. Looking at the results of this paper and Li et al. (2016), I cannot find enough support for the main conclusions in this paper.

1. P. 1, line 10-11, line 15; P. 2, line 4; P. 10, line 9-10, line 11-16

The emphasis on the prompt CHOCHO production under low NOx conditions is not explained well. Fig. 2 shows that the new GEOS-Chem mechanism has similar cumulative molar yields to MCM (although a little higher) for low- and high-NOx conditions. The increase of the yield at low NOx conditions is not higher than at high-NO conditions. Why is the yield increase at low-NOx conditions singled out? It is also unclear to me how in situ or satellite data can be used to separate prompt production of the GEOS-Chem mechanism from slower production of MCM at low NOx conditions when isoprene emissions are continuous over large regions in daytime. Tracking air parcels is impractical in this environment (see the later comments on section 3).

The much bigger problem is that Li et al. (2016) showed a factor 2-3 higher CHOCHO yields at low-NOx than high-NOx conditions, while the new GEOS-Chem mechanism and MCM have a factor of 3-4 lower yields at low-NOx than high-NOx conditions. A simple scaling of Fig. 1 by Li et al. and Fig. 2 of this paper gives a factor of 5-10 difference between the two studies at 0.01 ppbv NOx. This difference is much larger

than that between the new GEOS-Chem mechanism and MCM. If in situ and satellite observations can be used to constrain CHOCHO yields, this large difference between the two studies can surely be resolved.

Comparing Fig. 3 of this paper to Fig. 2 of Li et al., CHOCHO in this paper is close to 0 above 2 km while Li et al. showed CHOCHO concentrations within the range of the observations. Not looking at the details, one would think that the in situ observations suggest CHOCHO yields at low-NOx conditions are in line of Li et al. and are much higher than the new GEOS-CHEM mechanism or MCM. The 0-1 km data in Fig. 7 of this paper also suggest the model CHCHO yields can be higher at low-NOx conditions.

Fig. 7 will be more clear if the arithmetic NOx binning is changed to a logarithmic scale. 0-250 pptv covers both low- and mid- NOx conditions. Fig. 2 shows that CHOCHO cumulative yields do not change much for 0.5-1.5 ppb NOx, so it's not surprising that the changes of [CHOCHO]/[CHO] ratio in Fig. 7 are small. These are not "low" NOx conditions. I would not consider 200 ppt NOx as "low-NOx" either. A clear definition of low NOx is needed in the discussion. Fig. 2 shows that 200 ppt NOx, the cumulative CHOCHO yield is about 60% of 1 ppb NOx. I'd suggest adding a panel of the cumulative HCHO yield distribution in Fig. 2 to compare to CHOCHO.

HCHO has a background from CH4 oxidation. CHOCHO can have a background from oxidation of C2H2 but it is small and has a weak altitude dependence from 2 to 5 km. The observed CHOCHO decrease by a factor of 5 from 2 to 5 km in Fig. 3 does not look like a "background". I do not think that the unspecified instrument detection limit (line 20 on P. 6) can explain this type of altitude dependent decrease.

2. It is possible that the CHOCHO yields at low-NOx conditions are not the problem if simulated isoprene has large low biases. The suggestion of lacking shallow cumulus convection in the model (line 17-18 on P. 6) is a good reason to expect such a bias. Isoprene, MVK and MACR observations were used in section 3. Why are they not compared to model results in Fig. 3? PTRMS MVK+MACR data may have high biases.

[Figure]
Can WAS data be used to correct PTRMS data?

I suggest adding the comparisons of simulated isoprene, MVK+MACR, ozone, and CO to the observations in Fig. 3. It will be useful to see the spatial distributions of NOx, isoprene, MVK+MACR, and ozone in comparison to the observations, which Li et al. did not show. I suggest adding the model-observation comparisons of these species in Fig. 4.

3. P. 1 line 15; P. 2, line 1-4; P. 10, line 18-24

The OMI data used in section were June-August 2006-2007. Are the model simulations for the same period? The discussion in line 20-25 in section 4 (P. 9) seems to suggest that the model results are for the SENEX period. I think that GEOS-Chem results for June-August 2006-2007 are needed to support these rather tenuous conclusions.

Show Figs. 9 and 10 only for the high isoprene emitting SE region not the eastern US. The relatively high CHOCHO at 2-5 km is presumably due to isoprene oxidation unless one can show that VOCs other than isoprene (and its oxidation products) can produce that much CHOCHO at 2-5 km. There is no point of looking for this "background" CHOCHO over regions with low isoprene emissions.

The averaged model-OMI biases shown in Fig. 8 are not that large. How do these biases compare to retrieval uncertainties? OMI HCHO columns were increased by x1.67. What are the reasons? Why are CHOCHO retrievals not affected as HCHO retrievals?

4. Section 3

I do not think the parcel model analysis can be published. Below 1 km, air mass is actively mixed with continuous emissions in daytime over the Southeast. The assumption of air parcels isolated from emissions, i.e., Eqs (1) and (2), cannot be justified. The concept of "initial" isoprene is inappropriate in this context. Observed CHOCHO below 1 km is the result of oxidation of isoprene continuously emitted during an integrated

time period.

---

## Referee Comment (RC3) · Anonymous Referee #3 · 9 Jan 2017

This paper presents a new chemical mechanism for glyoxal (CHOCHO) production from isoprene oxidation that is used in the GEOS-Chem global chemical transport model. The glyoxal and formaldehyde (HCHO) yields from this mechanism are compared to those of the Leed's Master Chemical Mechanism (MCM v3.3.1) under different NOx conditions. The performance of this mechanism is then evaluated using CHOCHO and HCHO observations from the NOAA SENEX campaign, as well as 2006-2007 retrievals of HCHO and CHOCHO from the NASA Ozone Monitoring Instrument (OMI). The later is the first validation exercise for the OMI CHOCHO retrieval.

[Figure]

This is a well-written paper on an important topic in atmospheric chemistry, specifically the oxidation chemistry of isoprene and the ability to use satellite observations to infer isoprene emissions in important regions such as the southeast US. The methods seem reasonable and are described well, and the conclusions are generally supported by the results. All of my comments detailed below are minor or technical in nature, so I recommend publication after minor revisions to address them.

Minor Comments:

P2, L21: HOx is usually defined as OH + HO2, not plus all peroxy radicals, right? Why are you including organic peroxy radicals here?

P4, L5: There is no 2013 NEI – Do you mean the 2011 NEI with growth/control factors applied to simulate 2013?

P5, L2: This sentence is rally a conclusion, and so is out of place here. I'd suggest rephrasing to say that you explore if this pathway is consistent with SENEX observations of CHOCHO production in low NOx conditions in Section 3.

P6, L18-19: Do you have any evidence from more conserved species, like CO or aerosols, that vertical transport is underestimated?

P6, L24: It's not clear what you mean by "correlative analysis in the SENEX observations offer no insight." What analyses did you attempt?

P7, L1-2: I can see the NOx sensitivity in the GEOS-Chem plot in Figure 5 (perpendicular to the regression line), but I can't see it in the observations. Am I missing something?

Typos:

P3, L32: need a space before "Travis"

P5, L12: Expand "DSMACC".

---

## Author Comment (AC1) · 24 Mar 2017

We thank Jingqiu for referring us to Li et al. (2016). We have included a more detailed discussion between the chemical mechanisms from our work and Li et al. (2016) in Section 2.2 of the revised manuscript (Page 5, Line 5).
* * *

---

## Author Comment (AC2) · 24 Mar 2017

**1 Response to Reviewer 1**

**1.1 General Comments**

> **Comment 1**
>
> The authors present a studying using GEOS-Chem and box model simulations to understand and interpret observations of glyoxal and formaldehyde from the SENEX aircraft campaign over the Eastern US. In addition, they compare results to satellited-erived formaldehyde and glyoxal columns to determine if there is separate information about isoprene emissions that can be obtained from each species (which they do not find). The manuscript is well written and should be published after addressing the following minor comments.

**Response**

We thank the reviewer for their comments for improving our manuscript. Responses to specific comments are given below.

**1.2 Specific Comments**

> **Comment 2**
>
> Beta vs delta isoprene RO2 isomers: Can you clarify the yield of beta vs delta RIO2 isomers separately from the RIO2+NO ISOPN yield from each isomer? Table S1 indicates the authors recommend an update to the RIO2+NO → ISOPNB and ISOPND yields. I wonder how much of that update is due to the different isomer distribution and how much is due to the yield of ISOPN from each isomer. Note that in older versions of the isoprene chemistry, the yield of beta-RIO2+NO to produce ISOPNB and deltaRIO2+NO to produce ISOPND were different. Fisher et al., 2016 updated them to both be 9%. The 10% yield of delta isomers indicated in Figure 1 is higher than MCM (3.4%, page 5 line 29).

**Response**

The finalized isoprene organic nitrate yield (ISOPNB+ISOPND, both 9%) from Fisher et al. (2016) was not pushed to the simulation shown here, but should be included in the revised mechanism to properly simulate isoprene-derived organic nitrates. The scaling in our paper preserved the organic nitrate yields from the beta and delta pathways from the original mechanism (6.2% and 10%). This leads to a slight decrease in the MVK and HC5 yields ( 1.7% and 0.2% respectively). Based on Figure 1 of our paper, this will lead to a minor decrease in CHOCHO production ($\sim 1.2\%$ over the southeast US).

To avoid confusion about recommendations, we have included the Fisher et al. (2016) in the revised mechanism, and added a footnote to the table with the reaction used here.

**Comment 3**

The Li et al. paper is only cited twice despite using a very similar data set and working on a similar issue. More synthesis of results in the context of Li et al. would be helpful. For example, do both models agree in terms of the role of RO2 isomerization and its contribution to glyoxal? Comparing Figure 1 of Li et al. to Figure 1 in this paper indicates Li et al. predict a much larger role for RO2+HO2 relative to isomerization in producing glyoxal (but the figures are not directly comparable, so it is not clear).

**Response**

We now include more discussion about the differences between the GEOS-Chem and AM3 chemical mechanisms (Section 2.2). The justifications for our particular choices of CHO-CHO precursor yields are further expanded on in the supplementary material (Sections S1-3). Section S2 addresses the difference in RO2+HO2. Our CHOCHO yield via this channel is approximately 3 times lower than in AM3, due to differences in the yield from IEPOX. Our CHOCHO yields via ISOPO2 isomerization are similar (Section S3) however the pathways generating CHOCHO are different (DHDC photolysis in GEOS-Chem vs. HPALD photolysis in AM3). To our knowledge there is currently no obvious mechanism for HPALD photolysis to produce CHOCHO, and no details were provided in the paper cited by Li et al. (Stavrakou et al. (2010)).

**Comment 4**

Page 5, line 6-7. How was it determined that the model was not sensitive to aerosol reactive uptake? Was that through a simulation or estimated lifetime against uptake? The authors note that a background/free tropospheric source of glyoxal may be missing from the model. Have the authors considered whether or not reversible uptake of glyoxal, particularly if it is formed in the boundary layer and repartitions to glyoxal in the free troposphere, may provide this missing source?

**Response**

Originally this was based on a sensitivity simulation with and without CHOCHO aerosol uptake, using a reactive uptake coefficient of $10^{-3}$ (average of the Li et al. rates) at the OMI overpass time (13-14 pm local). However, this time period is where the OH and photolysis sinks should be strongest, and will overestimate their importance at times with less light.

In the revised version we have estimated the potential impact of aerosol uptake in the model via a steady state assumption (Section S4) that should resolve this shortcoming. The impacts of aerosols on the mean are now discussed in Section 2.2 (**P5,L17**).

> Li et al. (2016) found that CHOCHO concentrations are sensitive to aerosol reactive uptake. Our standard model simulation does not include this uptake, but we conducted a sensitivity simulation with a reactive uptake coefficient $\gamma = 2 \times 10^{-3}$ from Li et al. (2016). We find that CHOCHO concentrations decrease by only 10% on average (Section S4) because competing CHOCHO sinks from reaction with OH and photolysis are fast.

> ### Comment 5
>
> Page 8, near line 5 and Figure 7: The model shows a population of points with R(GF) < 0.01 while observations do not indicate such low R(GF) at any time. Do you know the cause of these low modeled R(GF)?

**Response**

The small number of low model $R_{GF}$ values may be due to a missing background source, similar to that missing in the free troposphere. Monoterpenes are a potential candidate, as MCM predicts that they produce CHOCHO in high yield. We have added this to the discussion of Figure 7 (**P8,L1**)

> Figure 6 also shows that there are is a small subset of points in GEOS-Chem with RGF values less than 0.01, reflecting low CHOCHO values in the model that are not found in the observations where the concentration floor is 0.05 ppbv (Figure 5). There may be a CHOCHO background missing from the model, possibly contributed by monoterpenes; MCMv3.3.1 predicts that the total CHOCHO yield from common monterpenes is high (Kaiser et al., 2015), and that they produce CHOCHO over a 5 timescale of days (Figure S11).

> ### Comment 6
>
> 5. Page 9, line 26-27 indicates finer scale, more temporally resolved data may provide valuable glyoxal data from satellite? Are the authors hypothesizing that R(GF)s may be more variable? Can GEOS-Chem predictions be used to test that theory?

**Response**

That is the idea, however as shown in Figure 7, GEOS-Chem can capture the trend, but not the magnitude of the high $R_{GF}$ values associated with prompt low-NOx production. This may be for a variety of reasons. Perhaps the resolution does not capture the high-isoprene low-NOx conditions seen in the observations (the nominal spatial resolution of TEMPO will be $\sim 5$ times higher). A higher yield, or faster photolysis rate of DHDC will also lead to higher $R_{GF}$ values in the model. These are both uncertain, and we have done our best based on available literature to estimate them.

> ### Comment 7
>
> Figure 6: Is the influence of NOx due to the effect on RO2 branching or OH?

**Response**

It is a combination of the two. High NOx increases OH and therefore increases isoprene photochemical processing. Increased photochemical processing has two main impacts. The slope will increase from higher isoprene production, but will be partially offset by additional removal of CHOCHO and HCHO by higher OH concentrations. Since the CHOCHO yields in GEOS-Chem are approximately constant for the first few hours of OH exposure time ($t_{OH}$), the 3 times increase between the low- and high-NOx slopes probably suggests

that photochemical processing is more important.

Figure 6 has now been removed from the manuscript due to concerns from another reviewer.

**1.3   Technical Corrections**

**Comment 8**

Second sentence of abstract could be reworded as it is not clear if HCHO is also measureable from space via the same technique as glyoxal or not.

**Response**

The sentence has been amended to

> Like formaldehyde (HCHO), another VOC oxidation product, it is measurable from space by solar backscatter.

**Comment 9**

Page 3, line 12: ?is in better agreement? than Vrekoussis?

**Response**

We now cite the retrievals we are referring to in the text (**P3, L10**).

> Our recent CHOCHO retrieval from the OMI satellite instrument (Chan Miller et al., 2014) is in better agreement with surface observations of CHOCHO and $R_{GF}$ (Kaiser et al., 2015) compared to those from GOME-2 (Vrekoussis et al., 2010) and SCIAMACHY (Wittrock et al., 2006) as a result of improved background corrections and removal of NO2 interferences.

**Comment 10**

Page 4, line 15: delta "vs beta" branching ratio

**Response**

Fixed

**Comment 11**

Page 4, line 15: forms as "a" second-generation

**Response**

Fixed

---

## Author Comment (AC3) · 24 Mar 2017

**1 Response to Reviewer 2**

**1.1 General Comments**

> **Comment 1**
>
> The analyses of CHOCHO and HCHO in this paper have many interesting components, three models (GEOS-Chem, DSMACC, and a parcel model), two mechanisms (GEOSChem and MCM), SENEX in situ observations, and OMI retrievals. A casual reading would suggest it is a publishable paper. But in the more careful second-round reading, I found many problems. I cannot recommend publishing this paper in its present form. Substantial changes are required. This work was probably done during the same period as Li et al. (2016, Observational constraints on glyoxal production. . .). The publication of that paper makes it necessary that differences between the two papers are resolved in this paper. Very little was done in this paper. Many differences were not mentioned. For example, Li et al. (2016) showed some effects of aerosol loss of CHOCHO in the mixed layer. Their budget shows that aerosol loss is 26% of total CHOCHO loss in the boundary layer, which is quite significant. The justification for not including this loss given in line 4-8 on P. 5 did not provide either the details on aerosol loss modeling or the results. Many analyses in this paper are similar to Li et al. (2016), but the results are different. The implications of these differences were not considered in this paper. The omission of comparing the simulation of isoprene to the observations, which was done by Li et al., may be an indication that the submission of this paper was rushed. Looking at the results of this paper and Li et al. (2016), I cannot find enough support for the main conclusions in this paper.

**Response**

We thank the reviewer for their time reading the paper. To clarify the differences between Li et al. and our paper, we have added a thorough comparison of the differences in CHOCHO formation pathways from isoprene (Section 2.2). Responses to specific comments are given below.

**1.2   Specific Comments**

> ### Comment 2
>
> **P. 1, line 10-11, line 15; P. 2, line 4; P. 10, line 9-10, line 11-16**
> The emphasis on the prompt CHOCHO production under low NOx conditions is not
> explained well. Fig. 2 shows that the new GEOS-Chem mechanism has similar cumu-
> lative molar yields to MCM (although a little higher) for low- and high-NOx condi-
> tions. The increase of the yield at low NOx conditions is not higher than at high-NO
> conditions. Why is the yield increase at low-NOx conditions singled out? It is also un-
> clear to me how in situ or satellite data can be used to separate prompt production of
> the GEOS-Chem mechanism from slower production of MCM at low NOx conditions
> when isoprene emissions are continuous over large regions in daytime. Tracking air
> parcels is impractical in this environment (see the later comments on section 3).

**Response**

The comparison was intended to discuss the differences between GEOS-Chem and MCMv3.3.1.
For the quote in question

> In GEOS-Chem, by contrast, the CHOCHO and HCHO yields show opposite
> dependences on $NO_x$, implying that they could provide complementary infor-
> mation on isoprene emissions.

The comment does not imply we are tracking air parcels. The idea is that if the time- and
NOx-dependence of CHOCHO and HCHO production from isoprene only differed by a
scaling factor (as is approximately true in MCMv3.3.1), then the associated CHOCHO and
HCHO spatial distributions would also only differ by a scaling factor. Hence CHOCHO
would provide redundant information for an isoprene emissions inversion.

> ### Comment 3
>
> The much bigger problem is that Li et al. (2016) showed a factor 2-3 higher CHOCHO
> yields at low-NOx than high-NOx conditions, while the new GEOS-Chem mechanism
> and MCM have a factor of 3-4 lower yields at low-NOx than high-NOx conditions. A
> simple scaling of Fig. 1 by Li et al. and Fig. 2 of this paper gives a factor of 5-10
> difference between the two studies at 0.01 ppbv NOx. This difference is much larger
> than that between the new GEOS-Chem mechanism and MCM. If in situ and satellite
> observations can be used to constrain CHOCHO yields, this large difference between
> the two studies can surely be resolved.

**Response**

The minimum mixed layer observed NOx concentration (with concurrent CHOCHO obser-
vations) was 87 pptv. The in-situ observations therefore do not provide a constraint on the
differences between our mechanisms at NOx levels of 10 pptv.

> **Comment 4**
>
> Comparing Fig. 3 of this paper to Fig. 2 of Li et al., CHOCHO in this paper is close to 0 above 2 km while Li et al. showed CHOCHO concentrations within the range of the observations. Not looking at the details, one would think that the in situ observations suggest CHOCHO yields at low-NOx conditions are in line of Li et al. and are much higher than the new GEOS-CHEM mechanism or MCM. The 0-1 km data in Fig. 7 of this paper also suggest the model CHCHO yields can be higher at low-NOx conditions.

**Response**

Concentrations above 2 km in Li et al. are also close to zero, but appear to be larger because the horizontal axis starts at -50 pptv. Whilst the SENEX data may support a modest increase in prompt low-$NO_x$ CHOCHO formation in our mechanism, they do not appear to support the extremely high yields at low-NOx production shown in Li et al. (2016) (Figure 4 in Li et al. (2016) shows large differences between the binned average $R_{GF}$ for the lower $NO_x$ bins).

> **Comment 5**
>
> Fig. 7 will be more clear if the arithmetic NOx binning is changed to a logarithmic scale. 0-250 pptv covers both low- and mid- NOx conditions. Fig. 2 shows that CHO-CHO cumulative yields do not change much for 0.5-1.5 ppb NOx, so it's not surprising that the changes of [CHOCHO]/[CHO] ratio in Fig. 7 are small. These are not 'low" NOx conditions. I would not consider 200 ppt NOx as 'low-NOx' either. A clear definition of low NOx is needed in the discussion. Fig. 2 shows that 200 ppt NOx, the cumulative CHOCHO yield is about 60% of 1 ppb NOx. I'd suggest adding a panel of the cumulative HCHO yield distribution in Fig. 2 to compare to CHOCHO.

**Response**

There are not enough points at the values discussed for the suggested logarithmic binning to be robust (there are only 3 observations with NOx < 100 pptv). The CHOCHO yield in Figure 2 shows that it is higher in GEOS-Chem at low-$NO_x$. Also the cumulative HCHO yield distribution is already shown in Figure 2.

> **Comment 6**
>
> HCHO has a background from CH4 oxidation. CHOCHO can have a background from oxidation of C2H2 but it is small and has a weak altitude dependence from 2 to 5 km. The observed CHOCHO decrease by a factor of 5 from 2 to 5 km in Fig. 3 does not look like a 'background'. I do not think that the unspecified instrument detection limit (line 20 on P. 6) can explain this type of altitude dependent decrease.

**Response**

The section in question only refers to the observations above 3 km ( and thus does not include the steep decrease between 2-3 km). We have amended the text with the precision stated by

Kaiser et al. (2015) (**P7, L3**)

> The CHOCHO observations in the free troposphere ($> 3$ km) have to be treated
> with caution since they are below the reported instrument precision (32 pptv,
> Kaiser et al. (2015)).

**Comment 7**

2. It is possible that the CHOCHO yields at low-NOx conditions are not the problem if simulated isoprene has large low biases. The suggestion of lacking shallow cumulus convection in the model (line 17-18 on P. 6) is a good reason to expect such a bias. Isoprene, MVK and MACR observations were used in section 3. Why are they not compared to model results in Fig. 3? PTRMS MVK+MACR data may have high biases. Can WAS data be used to correct PTRMS data?

**Response**

We have added profile comparisons of isoprene, MVK+MACR, CO and O3 to the supplementary information. We do not see large low biases in simulated isoprene (Figure S8). Wolfe et al. (2016) show a detailed comparison between iWAS and PTRMS MVK+MACR data. iWAS observations are biased high relative to the PTRMS data, possibly due to larger inlet conversion of ISOPOOH, or production within the canisters, with the latter explanation deemed less likely.

**Comment 8**

I suggest adding the comparisons of simulated isoprene, MVK+MACR, ozone, and CO to the observations in Fig. 3. It will be useful to see the spatial distributions of NOx, isoprene, MVK+MACR, and ozone in comparison to the observations, which Li et al. did not show. I suggest adding the model-observation comparisons of these species in Fig. 4.

**Response**

Figure 4 already shows $NO_x$. We have added the comparisons of isoprene, MVK+MACR, O3 and CO to the supplementary material (Figure S8 and S9).

**Comment 9**

**3. P. 1 line 15; P. 2, line 1-4; P. 10, line 18-24**
The OMI data used in section were June-August 2006-2007. Are the model simulations for the same period? The discussion in line 20-25 in section 4 (P. 9) seems to suggest that the model results are for the SENEX period. I think that GEOS-Chem results for June-August 2006-2007 are needed to support these rather tenuous conclusions.

**Response**

The model simulations are also from June-August 2006-2007. This simulation is from the model as described in the main text, except that it was performed globally at $2° \times 2.5°$ resolution. We have amended the main text to make this clearer (**P8, L15**).

The OMI observations are compared to a GEOS-Chem simulation covering the same period, at $2° \times 2.5°$ horizontal resolution.
* * *
**Comment 10**

Show Figs. 9 and 10 only for the high isoprene emitting SE region not the eastern US. The relatively high CHOCHO at 2-5 km is presumably due to isoprene oxidation unless one can show that VOCs other than isoprene (and its oxidation products) can produce that much CHOCHO at 2-5 km. There is no point of looking for this "background" CHOCHO over regions with low isoprene emissions.
* * *
**Response**

Both the CHOCHO and HCHO retrievals derive offset corrections over specific target regions where the column's values are assumed known. As such, the absolute value of the columns is less robust than the relative differences between columns. Looking at the spatial correlation, including both the low-isoprene region and the SE US isoprene hotspot, provides a means to validate the difference in satellite and model backgrounds.
* * *
**Comment 11**

The averaged model-OMI biases shown in Fig. 8 are not that large. How do these biases compare to retrieval uncertainties? OMI HCHO columns were increased by x1.67. What are the reasons? Why are CHOCHO retrievals not affected as HCHO retrievals?
* * *
**Response**

The random uncertainties in the retrievals can be assesed from spectrum fitting residuals, and are negligible after the spatiotemporal averaging applied in Figure 8 (e.g. for CHOCHO these are less than $2 \times 10^{13}$ molecules cm$^{-2}$). A bottom up estimate of the retrieval precision is much more difficult. Figure 9 is an attempt to indirectly assess the retrieval precision, which we have clarified in the text (**P8, L28**).

Excellent agreement is found for HCHO, providing an independent test of the correction to the OMI HCHO retrieval inferred from the SEAC4RS data (Zhu et al., 2016). Since GEOS-Chem can also replicate the HCHO-CHOCHO correlation in the SENEX data, the simulated CHOCHO columns can be used to indirectly validate the OMI CHOCHO observations.

The HCHO scaling was based on a validation of OMI HCHO observations using SEAC[4]RS HCHO observations by Zhu et al. (2016). The reasons for the bias are presently unknown, and we do not claim that the CHOCHO retrievals are not subject to similar error sources.

Comment 12

4. Section 3 I do not think the parcel model analysis can be published. Below 1 km, air mass is actively mixed with continuous emissions in daytime over the Southeast. The assumption of air parcels isolated from emissions, i.e., Eqs (1) and (2), cannot be justified. The concept of "initial" isoprene is inappropriate in this context. Observed CHOCHO below 1 km is the result of oxidation of isoprene continuously emitted during an integrated time period.

**Response**

We have removed the comparison between initial isoprene and CHOCHO/HCHO based on the reviewers concerns. The parcel model is still used to derive the $t_{OH}$ values in Figure 6 of the revised manuscript, as low MACR+MVK/ISOP ratios should still be a qualitative indicator for OH titration.

---

## Author Comment (AC4) · 24 Mar 2017

**1   Response to Reviewer 3**

**1.1   General Comments**

> **Comment 1**
>
> This paper presents a new chemical mechanism for glyoxal (CHOCHO) production from isoprene oxidation that is used in the GEOS-Chem global chemical transport model. The glyoxal and formaldehyde (HCHO) yields from this mechanism are compared to those of the Leeds Master Chemical Mechanism (MCM v3.3.1) under different NOx conditions. The performance of this mechanism is then evaluated using CHOCHO and HCHO observations from the NOAA SENEX campaign, as well as 2006-2007 retrievals of HCHO and CHOCHO from the NASA Ozone Monitoring Instrument (OMI). The later is the first validation exercise for the OMI CHOCHO retrieval. This is a well-written paper on an important topic in atmospheric chemistry, specifically the oxidation chemistry of isoprene and the ability to use satellite observations to infer isoprene emissions in important regions such as the southeast US. The methods seem reasonable and are described well, and the conclusions are generally supported by the results. All of my comments detailed below are minor or technical in nature, so I recommend publication after minor revisions to address them.

**Response**

We thank the reviewer for their helpful comments. Our responses to their specific comments are shown below, including corresponding changes to the manuscript.

**1.2   Specific Comments**

> **Comment 2**
>
> P2, L21: HOx is usually defined as OH + HO2, not plus all peroxy radicals, right? Why are you including organic peroxy radicals here?

**Response**

We have corrected this in the revised version (**P2, L20**).

> Isoprene impacts air quality and climate as a precursor to ozone (Geng et al., 2011) and secondary organic aerosol (SOA) (Carlton et al., 2009), and also affects concentrations of hydrogen oxide radicals ($HO_x \equiv OH + HO_2$ )

> **Comment 3**
>
> P4, L5: There is no 2013 NEI ? Do you mean the 2011 NEI with growth/control factors applied to simulate 2013?

**Response**

The scaling is relative to the 2011 NEI. We have corrected the sentence (**P4, L6**).

> NOx emissions are as described by Travis et al. (2016) including a 50% decrease in the anthropogenic source relative to the 2011 National Emission Inventory of the U.S. Environmental Protection Agency.

**Comment 4**

P5, L2: This sentence is really a conclusion, and so is out of place here. I'd suggest rephrasing to say that you explore if this pathway is consistent with SENEX observations of CHOCHO production in low NOx conditions in Section 3.

**Response**

The sentence was intended to reflect the motivation for including this pathway (which is not in MCMv3.3.1). It was not in our original mechanism, but rather it was motivated by discrepancy made apparent by the SENEX observations. We have modified the wording to try and convey this (**P5, L2**)

> As shown below, we find that this pathway can explain SENEX observations of prompt CHOCHO production under low-$NO_x$ conditions.

**Comment 5**

P6, L18-19: Do you have any evidence from more conserved species, like CO or aerosols, that vertical transport is underestimated?

**Response**

We have included a profile of CO in the supplement (Figure S8), and have updated the main manuscript (**P7, L1**)

> Modeled CO concentrations are also negatively biased above the mixed layer (Figure S8), providing further support that convective transport is underestimated.

**Comment 6**

P6, L24: It?s not clear what you mean by ?correlative analysis in the SENEX observations offer no insight.? What analyses did you attempt?

**Response**

To test for any obvious influences, we looked at the correlation coefficients (and rank correlations for robustness) for observations above 3 km, between 10-17 hours local time, for all VOC species measured during the campaign. We have updated the sentence to reflect this (**P7, L8**).

> There could be a free tropospheric source missing in the model, but it is unclear what this source could be, and correlative analysis of observed free tropospheric

CHOCHO with other species measured in SENEX offer no insight ($r < 0.3$ for all observed VOCs).
* * *
**Comment 7**

P7, L1-2: I can see the NOx sensitivity in the GEOS-Chem plot in Figure 5 (perpendicular to the regression line), but I can't see it in the observations. Am I missing something?
* * *
**Response**

The relationship to NOx, HCHO and CHOCHO is clearer when looking at $R_{GF}$ (Figure 6 in the revised manuscript). We have changed the text in the revised manuscript (**P7, L24**).

> The strong correlation between CHOCHO and HCHO might suggest that they provide redundant information for constraining isoprene emissions. However, examination of Figure 5 indicates higher observed CHOCHO-to-HCHO ratios ($R_{GF}$) at low-$NO_x$ concentrations, not captured by GEOS-Chem.

There is much less scatter in the GEOS-Chem points in Figure 5 due to the fact that transport by turbulent eddys is parameterized as diffusion (which removes variability associated with isoprene photochemical processing).
* * *
**Comment 8**

Typos:
P3, L32: need a space before "Travis"
P5, L12: Expand "DSMACC"
P5, L20: I think you need a comma before tOH
P10, L3: I think you need to hyphenate "NOx-dependent"
* * *
**Response**

The typos have been fixed in the revised manuscript.

---

## Author Response (AR2)

**1   Response to Reviewer 2**

**1.1   General Comments**

> **Comment 1**
>
> The added supplement information is very clear that the AM3 mechanism may have larger yields for wrong reasons. The additional figures are also very helpful. The new supplement makes it a much better paper. There are still three issues I think the authors need to correct. The changes will not require much additional work.

**Response**

We thank the reviewer for their additional comments for improving our manuscript. Responses to specific comments are given below.

> **Comment 2**
>
> One of my review questions is why there is a strong emphasis on the higher yield under low-NOx condition. It is still unanswered. Since this study and Li et al. both suggested a higher yield under lower NOx conditions (for different reasons), I suppose there must be observation evidence requiring it. But one of the authors? responses stated ?there are only 3 observations with NOx < 100 pptv?. Then there were no observations under really low-NOx conditions. Why would two studies focus on an issue that has not observation support? Can the authors make a clear statement in the paper as to whether or not a higher yield under low-NOx condition (than MCM) is required by the observations? It needs to be stated in the paper?s conclusions. Note that Fig. 6 is not the evidence if most of the missing CHOCHO in the lowest NOx bin is from pinene oxidation. In Fig. 6, the observations do not show an increase R(GF) with NOx (if the lowest NOx bin is not included); in contrast, the model shows the increase for the whole NOx range. The statement in Line 31 (P. 7) ?In both the model and observations there is a subset of low-NOx points with higher RGF values (0.03-0.06)? is very misleading since the model did not simulate the observations of RGF>0.035, which are the majority of 0.03-0.06 data points.

GEOS-Chem predicts that 44% of isoprene is lost through the isomerization and $HO_2$ pathways during SENEX (Figure 1), indicating that CHOCHO yields via the low-NOx pathways should influence the observations. We have now explicitly stated the reasons for the underestimate in the CHOCHO yield from isoprene in MCMv3.3.1 in the conclusion (P10,L20).

> Underestimated CHOCHO yields from isoprene in MCMv3.3.1 can be explained by missing production via DHDC photolysis, and a lower $\delta$-$ISOPO_2$ equilibrium fraction (3.4% in MCMv3.3.1 vs. 10% in GEOS-Chem).

Although not seen in the binned averages, the observations in Figure 6 do show an increase in $R_{GF}$ with NOx at low $t_{OH}$, corresponding to OH titration. Under these specific conditions

(high isoprene, low OH levels) the isomerization pathway should be the dominant CHO-CHO source. This is because (1) The isomerization branching pathway should be higher under low NOx and (2) because CHOCHO production from the other precursors depends on OH. The fact that there is an $R_{GF}$ enhancement is therefore evidence for production from via ISOPO$_2$ isomerization (which is not included in MCMv3.3.1). Although GEOS-Chem does not replicate the $R_{GF}$ magnitude, what is relevant is the trend with $t_{OH}$ ( i.e. at lower NOx levels, when $t_{OH}$ decreases we see an enhancement in $R_{GF}$). There are a few possibilities why the model cannot capture the magnitude of $R_{GF}$ (1) The model cannot resolve the most extreme titration events. (2) The yield via isomerization, or the photolysis rate of DHDC is underestimated - these have both been estimated based on proxies from existing literature (Section S3), and are thus subject to uncertainty.
* * *
**Comment 3**

P. 6, Line 31-32, P.7 Line 1-2. If shallow convection is the reason, why does it not affect MVK+MACR in the observations (Fig. S8)? The model CO bias is larger at 3-4 km than 1-3 km. That cannot be an indication of shallow convection. I think it?s better to acknowledge that the reasons for measured CHOCHO at 2-3 km are not understood.
* * *
We have updated the text to better acknowledge the uncertainty for the cause of the transition layer enhancement (P6,L31)

> During SENEX the mixed layer was typically capped by a neutrally stable transition layer of shallow cumulus convection which extended up to 3 km (Wagner et al., 2015), which could suggest that the model underestimates transport via this mechanism. However, the model does not underestimate other isoprene oxidation products in the transition layer, such as MVK+methacrolein (Figure S8). Another possible source of CHOCHO in the transition layer is via heterogeneous aerosol oxidation (Volkamer et al., 2015). However, specific aerosol precursors that produce CHOCHO at yields required to match the SENEX observations are currently unknown (Kaiser et al., 2015).
* * *
**Comment 4**

In the response, the authors stated that "The HCHO scaling was based on a validation of OMI HCHO observations using SEAC4RS HCHO observations by Zhu et al. (2016). The reasons for the bias are presently unknown, and we do not claim that the CHOCHO retrievals are not subject to similar error sources." This is a fair statement and should be noted in the conclusions of this paper. It will then put a statement like "The HCHO satellite data are better validated"? in the appropriate context.
* * *
We have amended the conclusion in response to the reviewers suggestion (P10,L28).

> Recent validation of the HCHO satellite data revealed negative retrieval biases (Zhu et al., 2016), which can be corrected using spatially uniform scaling factors (as done in this study). Since similar biases may exist for the CHOCHO retrieval, the scaled HCHO data should at present be preferentially used as proxy for isoprene emission.

---

## Author Response (AR3)

**1 Response to Reviewer 2**

**1.1 General Comments**

> **Comment 1**
>
> Dear Authors,
> Thank you for your response to Reviewer 2's further comments. Most of the comments have been addressed adequately. I have some further comments about Figure 6. It appears that even with the higher yields (than MCM), GEOS-CHEM still cannot capture the magnitude of the high RGF values in the observed data. Regarding comment 2 from the reviewer, I agree that 1) page 7 line 31 should be modified to more accurately reflect the difference in observation and model results, 2) an explicit statement on whether a higher yield (than MCM) under low-NOx condition is required by the observations should be included in the manuscript (and further justifications can be included if necessary). These will improve the clarity of the manuscript. Once these are considered and addressed, the manuscript will be accepted for publication in ACP.
> Best, Sally

Thanks Sally,

We have ammended the discussion of Figure 6 to more accurately reflect the model/observation difference (P7,L33)

> The observations contain a subset of low-NOx points with higher RGF values (0.03-0.06). The model also produces a subset of enhanced RGF values under low-NOx conditions, although peak RGF values are lower than the observations. In both cases, the enhanced RGF values coincide with short OH exposure times, which are caused by OH titration by isoprene. The high RGF reflects the relatively faster production of CHOCHO than HCHO in the early stage of isoprene oxidation under low-NOx conditions as shown by Figure 2. The presence of that population in the observations provides support for fast glyoxal production from the isomerization pathway of isoprene oxidation (Figure 1) that is present in GEOS-Chem but not in MCMv3.3.1. The model may not capture the highest observed RGF values due to uncertainties in the yield of DHDC from isoprene and its photolysis rate, both of which have been estimated based on literature proxies (Section S3).

We have added an explicit statement about the impact of missing DHDC on the MCMv3.3.1 yield in the conclusions (P10,L16)

> Mixed layer ($< 1$ km) observations show a strong CHOCHO-HCHO relationship that is reproduced in GEOS-Chem and is remarkably consistent across all conditions except at very low NOx where the [CHOCHO]/[HCHO] ratio (RGF) can be unusually high. This reflects prompt formation of CHOCHO under low-NOx conditions, which was missing from MCMv3.3.1 and is now simulated in our updated GEOS-Chem mechanism by DHDC photolysis. A previous model

comparison to SENEX showed that MCMv3.3.1 underestimates the CHOCHO yield from isoprene (Li et al., 2016). Our work shows the missing DHDC production pathway can explain approximately 60% of this underestimate, with the remainder caused by an underestimate of the $\delta$-ISOPO2 branching ratio (3.4% in MCMv3.3.1 vs. 10% in GEOS-Chem).